# Is one annotation enough?

### A data-centric image classification benchmark for noisy and ambiguous label estimation

**Lars Schmarje**[1][*]    **Vasco Grossmann**[1]    **Claudius Zelenka**[1]    **Sabine Dippel**[2]    **Rainer Kiko**[3]
**Mariusz Oszust**[4]    **Matti Pastell**[6] **Jenny Stracke**[5]    **Anna Valros**[7]    **Nina Volkmann**[7]
**Reinhard Koch**[1]

[1]MIP, Kiel University    [2]ITT, Friedrich-Loeffler-Institut    [3]LOV, Sorbonne Université
[4]Rzeszow University of Technology    [5]ITW, University Bonn    [6]University of Helsinki
[7]Luke, Natural Resources Institute Finland    [8]WING, University of Veterinary Medicine Hannover

## Abstract

High-quality data is necessary for modern machine learning. However, the acquisition of such data is difficult due to noisy and ambiguous annotations of humans. The aggregation of such annotations to determine the label of an image leads to a lower data quality. We propose a data-centric image classification benchmark with ten real-world datasets and multiple annotations per image to allow researchers to investigate and quantify the impact of such data quality issues. With the benchmark we can study the impact of annotation costs and (semi-)supervised methods on the data quality for image classification by applying a novel methodology to a range of different algorithms and diverse datasets. Our benchmark uses a two-phase approach via a data label improvement method in the first phase and a fixed evaluation model in the second phase. Thereby, we give a measure for the relation between the input labeling effort and the performance of (semi-)supervised algorithms to enable a deeper insight into how labels should be created for effective model training. Across thousands of experiments, we show that one annotation is not enough and that the inclusion of multiple annotations allows for a better approximation of the real underlying class distribution. We identify that hard labels can not capture the ambiguity of the data and this might lead to the common issue of overconfident models. Based on the presented datasets, benchmarked methods, and analysis, we create multiple research opportunities for the future directed at the improvement of label noise estimation approaches, data annotation schemes, realistic (semi-)supervised learning, or more reliable image collection. [2]

## 1   Introduction

High-quality data is the fuel of modern machine learning and almost all models improve with higher quality data [8, 80, 50]. Therefore, such data are a key component for developing future techniques. The acquisition of a large amount of data is considered particularly challenging due to the participation of humans in the process. Their mistakes or subjective interpretations of annotation tasks can lead to *noisy* or *ambiguous* labels, respectively [54, 16, 61, 28, 52, 9, 18, 29]. Consequently, the labels suffer from heteroscedastic aleatoric uncertainty which means that the data contains inherent noise, which is class- or even sample-dependent and negatively affects the quality [14].

In Figure 1, we present the impact of this uncertainty on the class "cat" in the CIFAR-10 dataset [32]. While all images have the same ground truth label in CIFAR-10, humans created agreeing annotations

---

[*]Corresponding Author, las@informatik.uni-kiel.de

[2]The source code is available at `https://github.com/Emprime/dcic`.
The datasets are available at `https://doi.org/10.5281/zenodo.7152309`.

36th Conference on Neural Information Processing Systems (NeurIPS 2022) Track on Datasets and Benchmarks.

only with varying rates from four to 100 percent [54]. This means that individual annotations can be expected to be noisy as they diverge from the majority opinion. Furthermore, a majority vote across multiple annotations can not capture the ambiguity between different images. In some extreme cases (red borders), we even see a disagreeing majority vote across all annotators from the expected ground truth class. We raise the question if all images should be treated equally if human annotations show such varying certainties. Taking a *data-centric* perspective [47, 62, 45, 25], we investigate the data in contrast to only the model for answering this question. Specifically, we propose a data-centric image classification (DCIC) benchmark that indirectly measures a method's ability to identify noisy and ambiguous labels and correct them. DCIC consists of ten real-world datasets of different domains (see Figure 2) and multiple human annotations for each image. The benchmark focuses on a data-centric view of the image classification problem by separating the data quality improvement and the classification performance into two tasks.

The main structure of the benchmark is divided into a *Labeling* and an *Evaluation* phase (see Figure 3a) which is comparable to established Teacher-Student-Approaches [70, 36]. Using this denotation, the benchmarked method will, as a teacher, improve labels during the first phase. These are then benchmarked in the second phase by analyzing their quality as training input to a student model. Be aware that we do not allow a knowledge transfer from the second phase to the first phase.

In detail, during the Labeling phase, we use samples from the distribution of the above-mentioned annotations to get different realistic label estimates as an *initialization*. The task of the benchmarked method is to improve these estimates for better performance of an other classification model in the second phase. In that phase (Evaluation), the obtained labels are used as input for training a fixed model and its performance is measured on a testing subset of the original data. In contrast to common model-centric deep learning approaches (see Figure 3b), we can vary the initialization for the same method and better separate its performance from the data improvement. The fixed model is used for the evaluation to facilitate distinguishing between performance gains due to improved input data and better learning of the method itself.

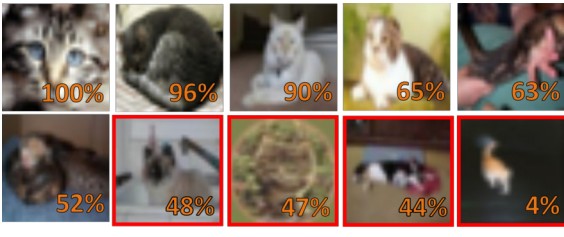

Figure 1: Are all images showing a cat? – Based on their ground truth labels from CIFAR-10 [32] they should all be cats. However, we give the agreement rate with the class cat from [54] in the lower right corner and see a wide range from four to 100%. Based on a majority vote, the last images (red border) would have not to be labeled as a cat but as dog, frog, dog, and deer, respectively. Based on these observations, we answer in our paper the question of whether all images should be treated equally as cats or if we should use multiple annotations and the resulting soft labels to capture this intrinsic noise and ambiguity.

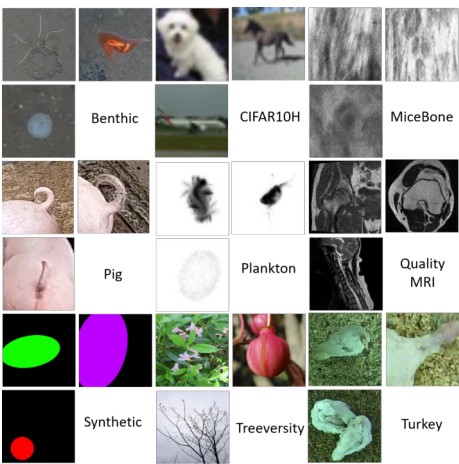

Figure 2: Three example images for all datasets. Details about the statistics of the dataset are given in Table 1.

Our benchmark is not only useful to evaluate existing methods, but will support research into algorithms for realistic datasets. Especially, it can bridge the research between semi-supervised learning and noise estimation based on realistic ambiguous noise patterns. We provide multiple algorithms as baselines and support the integration of more algorithms by common dataloaders for the two most popular deep learning frameworks: Tensorflow [1] and Pytorch [53]. We analyzed thousands of combinations of baseline methods, different initializations, and datasets. The obtained results confirm that the improvement of data quality leads to performance gains. Additionally, we investigated factors that influence the data quality and identified trends that lead to better learning of the underlying distribution.

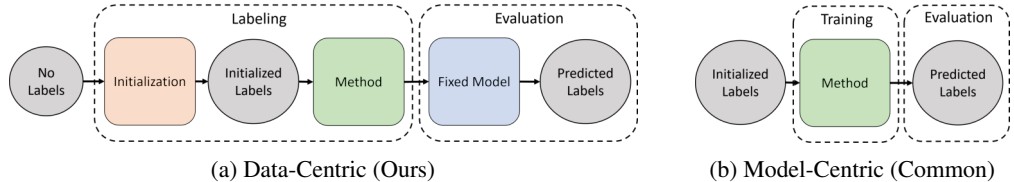

(a) Data-Centric (Ours)  (b) Model-Centric (Common)

Figure 3: Comparison of our data-centric approach with the commonly used model-centric approach. The circles and arrows represent the available label information in addition to the corresponding images. The squares represent the methods which generate / change these label. There are two main differences between our and the common model-centric approach. Firstly, we also look at how the raw unlabeled data is initialized and thus how many annotations are required. Secondly, we use a fixed model to evaluate the output of the benchmarked method. These differences lead to a greater separation of data quality and method performance on the final scores on the predicted labels.

Our key contributions are: (1) We collected and created ten real-world image classification datasets with multiple annotations per image. These annotations allow a realistic simulation of noise patterns and will be helpful for future research in machine learning on real world data sets. (2) We provide a multi-domain benchmark based on these datasets for noisy and ambiguous label estimation. We implemented 20 methods as comparison. The benchmark also covers the topic of cost and bridges research between semi-supervised learning and ambiguous label estimation. (3) We show that one annotation per image is not enough because model performance improves as more labels are given for each input. We identify that the current focus on hard labels for classifications is ill-suited to learn the underlying ground truth distribution. A change in data preprocessing especially in annotation protocols could mitigate this and lead to less overconfident models.

## 1.1 Related Work

Human annotations of one image can differ due to complex reasons. Next to mere individual errors, cognitive sciences have shown that human judgement under uncertainty is driven by a subjective bias and the context of the annotation process [66]. As labeling relies on human perception, data quality problems, including issues with noisy and ambiguous labels, have been broadly discussed in the literature [68, 4, 54, 77, 3]. While voting strategies have proven to be robust tools to remove outlying annotation errors in a single label scenario, they also eliminate subjective disagreement, even in cases with more than just one valid interpretation [56]. The impact of this information loss has been discussed in numerous studies, indicating limitations in capturing ground truth by just one label [71, 17, 22, 6]. We empirically support these arguments across 9 datasets and 20 methods.

As label noise can severely degrade the classification performance [50], learning with flawed training data has become a substantial field of research in which numerous strategies have been proposed: while sample selection methods separate clean and noisy data by evaluating small-loss or disagreement [78, 79], correction methods aim at relabeling wrongly assigned labels by either learning class prototypes [24] or by pseudo-labeling strategies that utilize confident predictions [38]. Multiple methods have been proposed, but are often evaluated on synthetic noise [44, 40]. However, Wei et al. showed that synthetic noise is different from real noise by humans, which limits the generality of findings [77]. Gao et al. proposed synthetic annotators with individual labeling behavior instead of random noise to reduce uncertainty in predictions [20]. We go beyond this by using human annotations to reproduce realistic noise pattern and we do not only look at annotation errors but also at ambiguous annotations from subjective interpretations.

Datasets like CIFAR-10H [54] and CIFAR-10N [77] address the problem of realistic noise by providing multiple annotations per image for example of the original CIFAR-10 dataset. By doing so, both publications demonstrate an improved performance and a higher robustness, while also claiming that further research in dealing with human noise is still inevitable. The utilization of soft label distributions instead of hard one-hot label encodings enables the detailed representation of subjective disagreement and improves the generalization with ambiguous datasets [54, 5, 34]. In our benchmark, we extend this idea to eight diverse datasets apart from CIFAR-10 for a broader evaluation.

If we want to use multiple annotations per image, we need to consider the cost of such annotations to make it feasible in a project. Current research such as semi-supervised learning [12, 67, 64, 63] could

be used to analyze only a portion of the data with multiple annotations. However, approaches which combine noisy labels with semi-supervised learning have not been extended to real-world image classification tasks or do not consider the possibility that one labels is not enough to capture the ambiguity of subjectivity [42, 81, 76]. We provide with our benchmark the datasets and infrastructure to bridge the research between semi-supervised learning and noise estimation.

Prediction uncertainty can be attributed to unexplainable noise in the given training or test dataset (aleatoric uncertainty) or a wrong model inference (epistemic uncertainty) and it can be difficult to approximate and differentiate between them [58, 2, 72, 30]. Several real-world noisy datasets have been utilized as a foundation for classification benchmarks [41, 39, 54], Song et al. provide a current survey on datasets and methods [68]. Moreover, most robust methods are evaluated based on the test set accuracy [40, 68, 77, 59, 82]. However, even a small change in the structure or parameters of a method can directly impact its performance, limiting the comparability [31]. Other fields, such as Bayesian Neural Networks, address this issue by comparing results to statistical simulations, for example [27]. A recent benchmark [49] tries to overcome this issue by providing a baseline for noisy labels as a form of uncertainty estimation [14]. However, this benchmark relies on synthetic noise or noisy datasets without knowledge about the underlying ground truth distributions [41]. We use a data-centric approach to minimize the impact of implementation detail differences and measure the impact of the data indirectly during the Labeling phase by evaluating on a fixed model.

## 2 Benchmark

Our benchmark is divided into two major phases: *Labeling* and *Evaluation*. In alignment with the Data-Centric Idea [47], we separate the improvement of the data (Labeling) from the improvement of the models (Evaluation). The benchmark can be utilized to analyze a variety of research questions, but we focus on evaluating methods that estimate noisy or ambiguous labels.

We use the terms *noisy* and *ambiguous* throughout this work synonymously because we often do not differentiate between their cause during the annotation process. As mentioned above, these causes are errors or mistakes of human annotators which can be recovered for noisy label or noise. Subjective interpretations, imprecise task descriptions or poor image quality lead to ambiguous labels.

In general, we have an image dataset $X$ with $k$ known classes and use human annotations to approximate the image labels. Each image $x \in X$ has an often unknown soft ground truth label $\hat{l_x} \in [0, 1]^k$. Therefore, we use $N$ hard human annotations $a_i \in \{0, 1\}^k$ with $i \in 1, ..., N$ as estimates of $\hat{l_x}$. We assume that an average of annotations ($l_x = \sum_{i=0}^{N} \frac{a_i}{N}$) is an approximation of this target label $\hat{l_x}$ as in [64]. Based on this definition, an annotation $a_i$ or hard label sampled from the distribution $l_x$ are in general of lower quality because they can not capture the aleatoric uncertainty of the soft label $l_x$. We split the data equally and randomly in five *folds* and ensure a similar class distribution between the folds as best as possible. For one run, we use three folds as training ($X_T$) and one fold each as validation ($X_V$) and test ($X_E$) data, respectively. We call such an assignment of folds to the training, validation and test data *slice*.

**Labeling** The Labeling phase consists of two steps. In the first step an initialization is used to get label estimates and in the second step, the benchmarked method $\Theta$ aims to improve these labels. As initialization, we acquire $m \in \mathbb{N}$ annotations for $n \in [0, 100]$ percent of the training and validation images.

We call the total number of required annotations *budget* $b = m \cdot n$ and report it as proportions of training and validation images ($|X_T \cup X_V|$). In general, a classification task gets easier with more annotations or a higher budget. Be aware that the same initialization results in the the same budget while the same budget can achieved by different initializations.

The used initialization schemes per method are defined later in this section. We chose fixed initialization schemes for better comparability between the methods. How these labels are improved by the method $\Theta$ is not restricted. However, annotations aside from the given initialization are not allowed to be used. Since we measure the quality by training a different fixed network in the next phase, a good label would be presumably as close as possible to $l_x$.

Table 1: Overview of the used datasets – # is an abbreviation for number. The class imbalance is given as the percentage of the smallest and largest class with regard to the complete dataset. The agreement is the percentage of annotations that agree with the majority vote. The scores ACC and $A\hat{C}C$ are given for the supervised baseline across three test folds. The access describes if the (raw) data is available openly, requires permission (restricted) or was not previously available (N/A). In the last column, datasets with modifications to the original data are marked with X. A modification might be adding more annotations or crop images to a region of interest.

| Name | # classes | Input size $[px]$ | # Images | Class Imbalance [%] | | Agreement [%] | # Annotations | ACC [%] | $A\hat{C}C$ [%] | Access | Updated |
|---|---|---|---|---|---|---|---|---|---|---|---|
| | | | | Smallest | Largest | Mean ± STD | Mean ± STD | Mean ± STD | Mean ± STD | | |
| Benthic | 10 | 112×112 | 4867 | 2.31 | 39.66 | 82.61 ± 19.67 | 4.54 ± 2.01 | 64.17 ± 0.63 | 83.36 ± 0.47 | Restricted | X |
| CIFAR-10H | 10 | 32×32 | 10000 | 9.88 | 10.16 | 95.44 ± 8.91 | 51.10 ± 1.54 | 90.75 ± 0.39 | 95.72 ± 0.12 | Open | |
| Mice Bone | 3 | 224×224 | 7240 | 14.75 | 70.48 | 85.06 ± 17.52 | 15.30 ± 21.90 | 61.88 ± 9.44 | 78.39 ± 1.95 | Restricted | X |
| Pig | 4 | 96×96 | 10237 | 7.82 | 41.23 | 65.32 ± 19.50 | 7.26 ± 2.29 | 35.97 ± 3.61 | 64.77 ± 0.79 | N/A | X |
| Plankton | 10 | 96×96 | 12280 | 4.16 | 30.37 | 93.26 ± 13.60 | 24.38 ± 44.17 | 89.89 ± 0.82 | 92.41 ± 0.41 | Restricted | |
| Quality MRI | 2 | 224×224 | 310 | 34.84 | 64.16 | 71.56 ± 12.27 | 99.94 ± 13.44 | 66.62 ± 3.55 | 75.81 ± 0.17 | Restricted | X |
| Synthetic | 6 | 224×224 | 15000 | 16.17 | 17.57 | 74.41 ± 24.28 | 98.86 ± 0.99 | 87.85 ± 0.48 | 74.65 ± 0.34 | N/A | X |
| Treeversity#1 | 6 | 224×224 | 9489 | 9.98 | 30.67 | 88.60 ± 16.13 | 14.78 ± 7.06 | 79.50 ± 1.53 | 89.20 ± 0.31 | Open | X |
| Treeversity#6 | 6 | 224×224 | 9826 | 8.77 | 31.26 | 66.53 ± 19.48 | 35.45 ± 11.47 | 56.71 ± 4.89 | 68.88 ± 0.72 | Open | X |
| Turkey | 3 | 192×192 | 8040 | 10.88 | 75.95 | 91.56 ± 13.82 | 14.85 ± 20.95 | 75.51 ± 2.80 | 86.89 ± 1.03 | Restricted | X |

**Evaluation**    In the Evaluation phase, the model and its hyperparameters are fixed to measure only the impact of the provided labels ($\Theta(x)$). The training of this fixed model $\Phi$ is calculated on the provided $\Theta(x)$ with $x \in X_T$. The best network parameters during training are selected based on a minimal divergence between $\Phi(x)$ and $\Theta(x)$ with $x \in X_V$. The generalization is then tested by measuring the difference between $\Phi(x)$ and $l_x$ for $x \in X_E$.

**Metrics**    Kullback-Leibler divergence ($KL$) [35] between $\Phi(x)$ and $l_x$ for $x \in X_E$ has been used as our main metric since it is an established method to measures the difference between two distributions [48]. We averaged in a 3-fold cross-validation per dataset for a high reproducibility. We used the three slices defined by $X_{V_i} = \{f_{i+1}\}$, $X_{E_i} = \{f_{i+2}\}$ and the rest as training ($X_{T_i} = \{f_i, f_{((i+2)\%5)+1}, f_{((i+3)\%5)+1}, \}$ with % for modulo) for the folds $f_1, ..., f_5$ with $i \in 1, 2, 3$ as the index of the slices. While $KL$ directly allows to measure the desired distribution divergence, we provide additional metrics as comparisons. We evaluate the accuracy ($ACC$) and F1-Score ($F1$) between $\Phi(x)$ and $l_x$ for $x \in X_E$ per class and report the mean across the classes, which is commonly called the macro value and allows evaluation even in the presence of class imbalance. We used the most likely class based on the evaluated distributions for these metrics. We analyze the calibration of the models by reporting the Expected Calibration Error ($ECE$) [23]. As reference, we provide all of these metrics also on the difference between the proposed label before the second training $\Theta(x)$ and the expected ground-truth $l_x$ for $x \in X_E$. The metrics are noted as $A\hat{C}C$, $\hat{F}1$ and $E\hat{C}E$. We report the Cohen's Kappa Score ($\kappa$) [46] as the measurement of the consistency of $\Theta(x)$ between the folds because more consistent labels result in higher model performance.

**Datasets**    We include ten real-world classification datasets in our benchmark. Since we need multiple annotations per image for the evaluation of the quality of labels and this information is often not available in existing datasets or insufficient for our benchmark, we collected, adopted, or extended annotations of the following datasets. Their details are shortly described below, while their properties and exemplary images are shown in Table 1 and Figure 2, respectively. As presented, the datasets vary across all reported properties, giving an opportunity to comprehensively evaluate considered methods. More challenging datasets are characterized by a high-class imbalance, a low average agreement, or a low number of annotations per image. Detailed reports about the collection process and remaining dataset specifics are given in the supplementary.

*1. Benthic* depicts images from the seafloor and consists of underwater flora and fauna. We used annotations from [65, 37] but filtered for at least three annotations per object and cropped the main image to this object. We combined classes with too few images in agreement with domain experts. *2. CIFAR-10H* is a variant of CIFAR-10 [32] introduced in [54]. Peterson et al. analyzed the underlying class distribution like us by reannotating the CIFAR-10 test set. In contrast to other variants like CIFAR-10N [77], this dataset provides more annotations per image. *3. MiceBone* consists of Second-Harmonic-Generation images of collagen fibers in mice [60]. The raw images were preprocessed as described in [64]. Since there is a need for multiple annotations per image, we hired workers to increase their number by a factor of five. *4. Pig* consists cropped tail images from European farms. The annotations were collected by hired workers with high domain knowledge. The goal is

Table 2: Overview of used methods grouped into supervised, semi-supervised and self-supervised. The second to fifth column describe if the method uses unlabeled data, makes noise estimation, what pretraining is the used input of the initialized dataset are hard or soft labels, respectively. The initialization schemes columns describe which schemes were evaluated for individual methods. The average runtime of the labeling phase is given in the last column.

| Name | Unlabeled Data | Noise Estimation | Pretraining | Labels | Initialization Schemes | | | | Avg. Runtime [h] |
|---|---|---|---|---|---|---|---|---|---|
| | | | | | SL | SL+ | SSL | SSL+ | |
| Baseline | | | | Soft | X | X | X | X | 0.00 |
| Heteroscedastic (Het) [15] | | X | | Hard | X | X | X | X | 0.50 |
| SNGP [43] | | X | | Hard | X | X | X | X | 0.29 |
| ELR+ [44] | | X | ImageNet | Hard | X | X | X | X | 0.09 |
| Mean-Teacher (Mean) [70] | X | | | Hard | X | | X | | 1.08 |
| Mean-Teacher (Mean+DC3) [64] | X | | | Hard | X | | X | | 1.20 |
| $\pi$-Model ($\pi$) [36] | X | | | Hard | X | | X | | 1.03 |
| $\pi$-Model ($\pi$+DC3) [64] | X | | | Hard | X | | X | | 1.15 |
| FixMatch [67] | X | | | Hard | X | | X | | 4.53 |
| FixMatch +DC3 [64] | X | | | Hard | X | | X | | 4.01 |
| Pseudo-Label (Pseudo v1) [38] | X | | | Hard | X | | X | | 1.10 |
| Pseudo-Label (Pseudo v1 +DC3) [64] | X | | | Hard | X | | X | | 1.40 |
| Pseudo-Label (Pseudo v2 hard) [38] | X | | ImageNet | Hard | X | X | X | X | 0.16 |
| Pseudo-Label (Pseudo v2 soft) [38] | X | | ImageNet | Soft | X | X | X | X | 0.12 |
| Pseudo-Label (Pseudo v2 not) [38] | X | | | Soft | X | X | X | X | 0.12 |
| DivideMix [40] | X | X | ImageNet | Hard | X | X | X | X | 1.39 |
| BYOL [21] | X | | Self-Supervised | Hard | X | | X | | 2.59 |
| MOCOv2 [13] | X | | Self-Supervised | Hard | X | | X | | 7.94 |
| SimCLR [11] | X | | Self-Supervised | Hard | X | | X | | 5.89 |
| SWAV [10] | X | | Self-Supervised | Hard | X | | X | | 4.17 |

the classification of the injury degree of the tail. *5. Plankton* is a collection of underwater plankton images with multiple annotations from citizen scientists [61]. We use the preprocessing described in [64]. *6. QualityMRI* consists of human magnetic resonance images (MRI) with a varying quality and multiple subjective quality ratings gathered in tests with radiologists. It was introduced and evaluated in [51, 69]. *7. Synthetic* dataset was generated for the purpose of this study. It consists of images that contain one blue, red, or green circle or ellipse on a black background. To create ambiguous images, we added color and axis interpolations of these classes. *8+9. TreeVersity* is a publicly available crowdsourced dataset of plant images from the Arnold Arboretum of Harvard University[3]. In the crowdsourcing project, the images were tagged with a given set of labels. We used a simplified version with six classes where we combined classes with too few images. Only images with at least three tags were used. Tags are not the same as class labels, therefore, we provide two subsets of TreeVersity. In TreeVersity#1, we filtered for exactly one given tag of the six possible ones per user which is similar to a classification. In TreeVersity#6, we filtered for a maximum of six different tags which means we did not apply any restrictions. *10. Turkey* is a dataset with images of turkeys and their injuries [74, 75]. We used the preprocessing described in [64] and extended the original annotations, increasing their number by a factor of five with hired workers.

**Methods** We compare a variety of recent supervised, self-supervised, and semi-supervised algorithms against our baseline. The baseline does not adjust the initialized dataset in the first phase in any way and just forwards these labels to the supervised training of the second phase. Thus it is equivalent to supervised learning in a model-centric benchmark. We selected the other methods based on their recency, access to authors code or reimplementations and if they are state-of-the-art or commonly used as comparisons in the literature. The *supervised* methods are Heteroscedastic [15], SNGP [43], and ELR+ [44]. The *semi-supervised* methods are Mean-Teacher [70], $\pi$-Model [36], FixMatch [67], DC3 [64], Pseudo-Label [38], and DivideMix [40]. The *self-supervised* methods are BYOL [21], MOCOv2 [13], SimCLR [11], and SWAV [10]. Detailed descriptions about most of them are given in [63] and their key characteristics are presented in Table 2. We use the reported hyperparameters for Imagenet [33] or Webvision [41] by the original authors to ensure a comparison out-of-the-box across different image domains. For DC3[64], we investigated the combinations with Mean-Teacher, $\pi$-Model, FixMatch, and Pseudo-Label. For Pseudo-Label, we used two different implementations (v1 and v2) and variants with or without pretraining and soft or hard labels as input. In total, this results in 20 investigated methods. For better referencing, we group them as described above but put methods that *use soft labels* into their own group.

---

[3]https://arboretum.harvard.edu/research/data-resources/

**Initialization Schemes** We investigated a fixed set of initialization schemes and note them $m$-$n$ for $m$ annotations for a subset of data with a relative size $n$. For easier reference, we group them as

- *Supervised Learning (SL)* 01-1.00
- *Supervised Learning+ (SL+)* 03-1.00, 05-1.00, 10-1.00
- *Semi-Supervised Learning (SSL)* 01-0.10, 01-0.20, 01-0.50
- *Semi-Supervised Learning (SSL+)* 10-0.10, 05-0.20, 02-0.50

**Implementation Details** The final results depend on a good set of fixed hyperparameters like learning rate for the model $\Phi$ for each dataset during the evaluation. Therefore, we determined them by applying Hyperopt [7] with 100 search trials across the same grid of parameters for all datasets. The target was the minimization of $KL$ between $\Phi(x)$ and $l_x$ for the baseline experiment with exactly ten annotations per image across one slice. We executed these and later experiments on an Nvidia RTX 3090 with 24GB VRAM or comparable hardware. Some combinations of models and input sizes could not fit on this hardware and therefore were ignored to keep the needed hardware to a minimum. Details about the used parameter grid are given in the supplementary. We ensure that all folds are randomly generated, while restrictions about similar images are considered. Without these restrictions, similar images, e.g., frames from the same camera might lead to an information leakage between the folds which would negatively influence the interpretability of the results.

## 3 Analysis

The evaluation was conducted across combinations of all datasets, methods, initialization schemes, and slices. A complete cross-combination would result in 5400 experiments from which we selected 3456 experiments to save resources since some combinations would not add more insights e.g. due to inferior performance of similar methods. A detailed overview of the initialization schemes used per method can be found in Table 2. As shown in Table 1, we have a large variability between the datasets, especially in $ACC$ and $A\hat{C}C$ that range from 36% to 96% for the baseline. Due to the fact that the baseline does not adjust the initialization, $A\hat{C}C$ can be seen as the performance of humans in improving the labels for the given budget. The datasets Benthic, MiceBone, Pig, Treeversity#6, and Turkey have an over 10% lower $ACC$ than the expected $A\hat{C}C$, which marks them as particularly challenging for the model. Moreover, the datasets Benthic, MiceBone, Pig, QualitMRI, and Treeveritsy#6 have an $A\hat{C}C$ of lower than 85% which marks them as difficult even for humans. The $A\hat{C}C$ of Synthetic is even lower due to the artificially created labels. Due to this variability, an average across the scores can be misleading. For this reason, we report the median in this paper and report the full results including the standard error of the mean (SEM) in the supplementary.

**What metrics should we use?** We analyzed the correlations between our metrics to determine which contain the same or similar information and which are complementary. All calculated Pearson correlation coefficients have a p-value $< 0.01$ and the four strongest correlations are $ACC$ vs. $F1$ (0.99), $KL$ vs. $ECE$ (0.68) $F1$ vs. $\kappa$ (0.77) and $ACC$ vs. $\kappa$ (0.77). The other correlations are around -0.5. Selected correlations are illustrated in Figure 4 and additional graphics and analysis are given in the supplementary. $F1$ balances the precision and recall but in our experiments we see almost identical values to $ACC$ which we credit to the averaging per class. This means we only need to concentrate on $ACC$ as a classification score. Overconfident models are a problem of modern machine learning [23] and the higher correlation between $KL$ and $ECE$ compared to any of the two with $ACC$, $F1$ or $\kappa$ indicates that our focus on classification metrics like $ACC$ and $F1$ could be the issue. Only metrics like $KL$ and $ECE$ consider the complete distribution and which justifies using mainly $KL$ for the evaluation of this benchmark.

**One annotation is not enough** It is to be expected that more and better data should lead to increased performance which we quantify in Figure 5a. It can be seen that all metrics improve with an increased budget in the form of more annotated data or more annotations per image. However, the impact is lower for more annotations per image. For example, $ACC$ increases from around 55% to 69% for the full supervision. Up to 10 annotations per image increase the score only to around 72%. This difference can be explained by the fact that additional annotations are most valuable to improve uncertain labels. In alignment with previous research [71, 22, 6], these results across thousands of

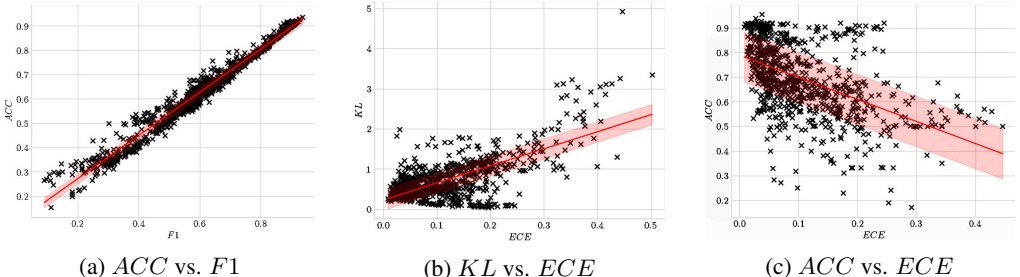

(a) $ACC$ vs. $F1$        (b) $KL$ vs. $ECE$        (c) $ACC$ vs. $ECE$

Figure 4: Correlations between selected metrics across all experiments. The red line represents the linear regression between the metrics and the light red area the mean absolute error of the regression.

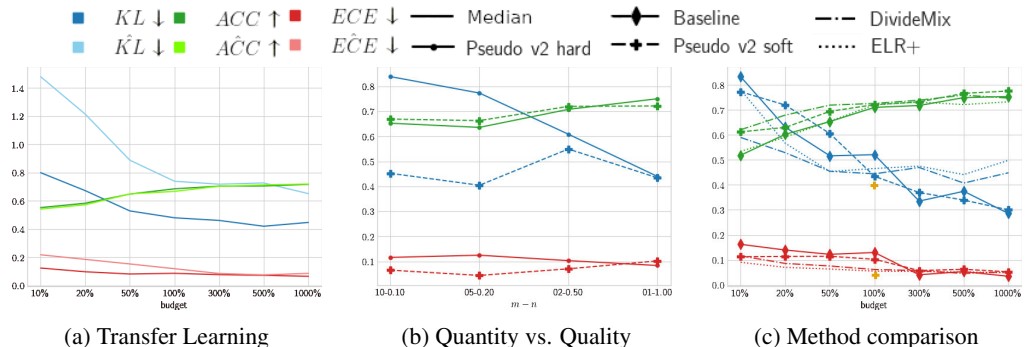

(a) Transfer Learning      (b) Quantity vs. Quality      (c) Method comparison

Figure 5: Analysis of all or selected methods across different budgets or initialization schemes. For details about the definitions see section 2. The orange crosses in (c) represent the best performance of Pseudo v2 soft with another initialization scheme see (b). The budget is increased by an raised portion of labeled data ($n$) until 1.00 and then increased further by using additionally multiple annotations per image ($m$).

experiments empirically justify that one annotation is not enough to capture the ground-truth of an item. Some improvements can be gained from correction annotation errors via a majority vote but the high disagreement and low $A\hat{C}C$ of the baseline on some datasets support the hypothesis that ambiguous annotations are a main source for the improvement. This ambiguity can not be described with a single hard majority vote and thus highlights the importance of using soft labels. Additionally, we see that $KL$ is about half as small as $\hat{KL}$, $ECE$ is around 3-10% lower than $E\hat{C}E$ and $ACC$ is 1-2% better than $A\hat{C}C$. As shown previously by Hinton et al. [26, 12] the knowledge distillation via a neural network into soft labels can be beneficial for $ACC$. We find the impact for metrics like $KL$ and $ECE$ even higher which supports our design of a two-phase benchmark.

**Limits of current state-of-the-art** To determine the best-performing state-of-the-art method, we gathered their relative improvements over the baseline in Table 3a. The best algorithm for each type of the soft, semi-supervised, supervised, self-supervised approaches based on the average performance across the budgets of 10%, 100%, and 1000% are Pseudo v2 soft, DivideMix, ELR+, and Mocov2, respectively.We visualize the best three of them in Figure 5c and give detailed results across the datasets for the budget 100% in Table 3b. The full results can be found in the supplementary. All top three methods are pretrained on ImageNet and outperform the rest in the field they were designed for. DivideMix is the best during partial supervision (budget < 100%), ELR+ is more noise robust (budget > 100%), and Pseudo v2 soft has the lowest $KL$ score (budget > 100%). It is important to note that ImageNet pretraining leads to improvements on many datasets (see Pseudo v2 not in the supplementary) but also to worse results on others. It needs to be investigated if other pretraining such as CLIP [55] or unsupervised pretraining on larger datasets [12, 57] could improve on these results. Overall, the current state-of-the-art methods are insufficient for a label preprocessing across all domains. For a high budget any investigated method is worse than the supervised baseline without adjustments of the initialized dataset which shows the lack of appropriate algorithms for such budgets.

Table 3: Results for the best performing methods – The best metric is marked bold while the 2nd and 3rd best are italic. Only methods with at least one top3 ranking across the budgets are presented. The full results are in the supplementary. (a) show the relative improvement over the baseline. (b) are detailed results for the budget of 100% across all datasets.

(a) Improvements

| Budget | 10% | | 100% | | 1000% | |
|---|---|---|---|---|---|---|
| | Median | Mean ± SEM | Median | Mean ± SEM | Median | Mean ± SEM |
| ELR+ | -0.16 | -0.59 ± 0.22 | **-0.13** | -0.17 ± 0.06 | *0.15* | *0.24 ± 0.06* |
| SGNP | -0.26 | -0.59 ± 0.27 | 0.00 | -0.15 ± 0.08 | 0.20 | 0.26 ± 0.04 |
| DivideMix | -0.21 | **-0.78 ± 0.25** | -0.03 | -0.17 ± 0.08 | *0.16* | 0.31 ± 0.07 |
| Mean | -0.14 | -0.59 ± 0.22 | *-0.12* | **-0.22 ± 0.08** | N/A | |
| π | **-0.33** | -0.63 ± 0.19 | -0.06 | -0.17 ± 0.05 | N/A | |
| π+DC3 | *-0.32* | -0.62 ± 0.20 | *-0.10* | **-0.22 ± 0.06** | N/A | |
| Pseudo v2 hard | *-0.30* | -0.39 ± 0.16 | -0.03 | *-0.19 ± 0.08* | 0.19 | 0.28 ± 0.04 |
| Pseudo v2 soft | -0.28 | -0.40 ± 0.19 | -0.04 | -0.17 ± 0.06 | **0.01** | **0.00 ± 0.02** |
| MOCOv2 | -0.29 | *-0.63 ± 0.22* | -0.08 | -0.13 ± 0.10 | N/A | |

(b) Details 100% Budget

| Dataset | Benthic | CIFAR10H | MiceBone | Pig | Plankton | QualityMRI | Synthetic | Treeversity#1 | Treeversity#6 | Turkey |
|---|---|---|---|---|---|---|---|---|---|---|
| Baseline | 1.17 ± 0.04 | 0.41 ± 0.02 | 0.55 ± 0.06 | 0.75 ± 0.05 | 0.34 ± 0.02 | 1.73 ± 0.48 | **0.08 ± 0.01** | 0.49 ± 0.04 | 1.02 ± 0.03 | *0.40 ± 0.08* |
| ELR+ | **0.70 ± 0.01** | 0.29 ± 0.02 | **0.29 ± 0.01** | 0.62 ± 0.09 | **0.24 ± 0.03** | 1.44 ± 0.83 | 0.18 ± 0.02 | **0.46 ± 0.01** | **0.47 ± 0.05** | 0.52 ± 0.05 |
| SGNP | 1.11 ± 0.05 | 0.38 ± 0.01 | 0.56 ± 0.14 | 0.77 ± 0.07 | 0.33 ± 0.02 | **0.25 ± 0.14** | 0.10 ± 0.00 | **0.46 ± 0.01** | 1.07 ± 0.05 | 0.42 ± 0.08 |
| DivideMix | 0.87 ± 0.07 | 0.36 ± 0.02 | *0.38 ± 0.05* | 0.95 ± 0.04 | 0.34 ± 0.01 | 0.46 ± 0.26 | 0.33 ± 0.00 | 0.47 ± 0.01 | 0.62 ± 0.05 | 0.43 ± 0.08 |
| Mean | 0.80 ± 0.06 | **0.28 ± 0.02** | *0.38 ± 0.02* | 0.64 ± 0.03 | 0.32 ± 0.01 | *0.35 ± 0.11* | 0.09 ± 0.01 | 0.61 ± 0.01 | 0.52 ± 0.03 | 0.75 ± 0.07 |
| π | 0.71 ± 0.03 | 0.33 ± 0.02 | 0.38 ± 0.00 | 0.83 ± 0.18 | 0.30 ± 0.02 | 0.98 ± 0.04 | **0.08 ± 0.01** | 0.52 ± 0.01 | *0.51 ± 0.03* | 0.55 ± 0.01 |
| π+DC3 | 0.72 ± 0.06 | 0.30 ± 0.02 | 0.39 ± 0.04 | 0.67 ± 0.06 | 0.29 ± 0.03 | 0.88 ± 0.34 | **0.08 ± 0.00** | 0.48 ± 0.01 | *0.49 ± 0.00* | 0.44 ± 0.08 |
| Pseudo v2 hard | 0.97 ± 0.10 | 0.43 ± 0.02 | 0.45 ± 0.10 | 0.67 ± 0.04 | 0.31 ± 0.03 | 0.29 ± 0.05 | 0.10 ± 0.01 | 0.46 ± 0.03 | 0.99 ± 0.09 | 0.39 ± 0.05 |
| Pseudo v2 soft | 1.00 ± 0.08 | 0.41 ± 0.02 | 0.40 ± 0.08 | 0.70 ± 0.06 | 0.32 ± 0.06 | 0.62 ± 0.16 | 0.10 ± 0.01 | **0.46 ± 0.01** | 0.83 ± 0.13 | **0.37 ± 0.08** |
| MOCOv2 | 0.91 ± 0.05 | 0.98 ± 0.02 | *0.37 ± 0.04* | **0.56 ± 0.01** | 0.52 ± 0.01 | 0.29 ± 0.11 | 0.13 ± 0.01 | 0.80 ± 0.03 | 0.61 ± 0.01 | 0.42 ± 0.08 |

Moreover, an average better performance does not mean that the gains are equal across all datasets. For example, ELR+ has the lowest KL at a budget of 100% for five out of ten datasets but on the QualityMRI dataset, it is among the worst methods. This means while some methods might work on some datasets they might not generalize to other datasets. Overall, we see the highest $KL$ for the datasets Benthic, Pig, Quality MRI and Treeversity#6 which also have the lowest agreement as seen in Table 1 except for the synthetic dataset. These datasets also show the largest variance in results across the methods. We conclude that the impact of the data is larger than the impact of the current preprocessing of state-of-the-art methods. This highlights the importance for investigating the data and label generation more if they a more impactful than the method itself.

While a higher budget leads to improved metrics, it also matters how it is used. In Figure 5b, we investigated the impact on $KL$ and $ACC$ for a budget of 100% for Pseudo-Label using hard or soft labels. We find that the accuracy is comparable between the methods and increases with a rising percentage of labeled data ($m$). For hard labels, the $KL$ improves equally. If we use soft labels for training, we see lower results for 05-0.20 and 10-0.10. We conclude that we should investigate more how we distribute our budget if increasing it is not an option.

## 4 Discussion

Overall, we can confirm several previous research hypotheses while identifying missing information and thus new research opportunities with our novel datasets and benchmark.

We can demonstrate that data quality positively impacts the classifications scores like $ACC$ and $F1$ and distribution-based scores like $KL$ and $ECE$. Knowledge distillation can improve the approximation of the underlying distribution further. We agree with previous research [71, 17, 22, 6] that one annotation is not enough and we need to use soft labels to handle ambiguous data. $KL$ and $ECE$ are highly correlated (0.7) and are improved more when using soft labels. We believe that focusing on learning the real distribution and thus minimizing $KL$ can lead to less overconfident models. Using soft labels as input seems to be crucial for achieving this since hard labels and classification metrics like $ACC$ lead to models which slightly ignore the real ground truth distribution.

Nevertheless, most of the investigated state-of-the-art method do not use soft labels and often interpret noise only as errors in the annotation process. These issues need to be addressed in future research and a simple method like Pseudo v2 soft illustrates how the $KL$ score can be lowered with this approach. For the largest budget, the baseline is the best model and even special noise estimation algorithms like ELR+ [44] and SNGP [43] can not achieve better results. We see a high variance across the datasets for different methods in our benchmark. However, we need methods which work across a variety of domains out-of-the-box to allow an easy application to current research question in other domains. Another practical issue is that we need to find solutions for acquiring soft labels even with a limited budget. In many research projects, it is difficult to annotate thousands of images with domain experts and annotating them multiple times would only increase the costs further. Thus, we need to bridge the research in semi-supervised learning and ambiguous and noise estimation. Such combinations could allow the usage of soft labels on a subset of images and simultaneously determine annotation errors. Our benchmark and datasets allowed the identification of these issues and thus could also be used to research new methods to solve these issues. We are confident that our benchmark and datasets can facilitate the bridged research on the topic of semi-supervised learning and ambiguous image estimation for real world image classification problems.

**Impact**   This work as a benchmark provides ten datasets and a detailed evaluation across 20 algorithms on this benchmark. The provided data can allow the investigation of research questions on the topics of e.g. noise estimation, data annotation scheme, or realistic semi-supervised learning. This work is intended to allow and spark future research and thus no direct social impacts are expected. However, this basic research is time and resource-consuming. For the final evaluation, we conducted experiments with about 5500 GPU hours which equals around 600kg $CO_2$. For this reason, we limited the evaluation always to necessary elements when possible in order to not increase the needed GPU hours further.

**Limitations**   The 20 investigated algorithms are only evaluated with one fixed set of hyperparameters across different datasets during the labeling phase. For optimal performance, a tuning per algorithm would have been required. We were interested in the general performance out-of-the-box and therefore neglected this issue due to resource minimization. The researched datasets are all below 15,000 images and the unsupervised learning potential on millions of images could not be investigated. We want to provide a detailed analysis in relation to the underlying distribution $l_x$ per image which is only possible with multiple annotations per image. For larger datasets, this effort was just not feasible. Classification with hundreds or more classes are also not feasible because the annotation costs increase with the number of classes. As described above we conducted more than a thousand experiments but we had to select and combine several results in a comprehensive manner in this paper. These aggregations can not capture all details. Much more detailed analysis e.g., per dataset would be possible and thus we included all raw results in the supplementary. Due to the fixed initialization scheme, we can not investigate active learning approaches. However, this restriction is chosen to allow a better comparability and future researchers could decide against such a restriction.

**Conclusion**   In alignment with previous research, we show that one annotation is not enough to handle ambiguous and noisy images and their underlying ground truth distribution. Multiple annotations and some kind of soft label are required to capture the difference in the images. Future research needs to investigate in more detail how annotations are being created, including annotation costs. We show that current state-of-the-art can help under certain budget or dataset constraints. However, methods with consistent results across a variety of datasets and budgets are missing. We release all datasets and the benchmark publicly to enrich future research on these topics.

## Acknowledgments and Disclosure of Funding

We thank Mark Collier for his valuable feedback and discussion about the benchmark. We thank the annotators Kristina Ahlqvist, Daniel Grundig, Stine Heindorff, Jana Krambeck, Kathrin Körner, Richard Lange, Miina Tuominen-Brinkas and Emirhan Ustalar for their valuable work.

We acknowledge funding of L. Schmarje by the ARTEMIS project (Grant number 01EC1908E) funded by the Federal Ministry of Education and Research (BMBF, Germany). R. Kiko also acknowledges support via a "Make Our Planet Great Again" grant of the French National Research Agency within the "Programme d'Investissements d'Avenir"; reference "ANR-19-MPGA-0012". V. Grossmann is employed with funds provided by Kiel Marine Science (KMS) and Future Ocean Network (FON) by Kiel University. Funds to conduct the PlanktonID project were granted to R. Kiko and R. Koch (CP1733) by the Cluster of Excellence 80 "Future Ocean" within the framework of the Excellence Initiative by the Deutsche Forschungsgemeinschaft (DFG) on behalf of the German federal and state governments. Turkey data set was collected as part of the project "RedAlert – detection of pecking injuries in turkeys using neural networks" which was supported by the "Animal Welfare Innovation Award" of the "Initiative Tierwohl".

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
