# OpenReview forum: "Is one annotation enough? -  A data-centric image classification benchmark for noisy and ambiguous label estimation"
_NeurIPS.cc/2022/Track/Datasets_and_Benchmarks — NeurIPS 2022 Datasets and Benchmarks _

### Official Review · Reviewer_otHK · 2022-07-13
**Effect of annotation budget on machine learning**

**Rating:** 4
**Confidence:** 4
**Correctness:** Yes.

**Strengths:**

There are few examples of datasets where all annotations are given, e.g. Cifar10H only provides multiple annotations on the test set.



**Weaknesses:**

row 99 => section 2 within section 2.

Few statistics about the data are given (variance, median, voting disagreement, etc.)

One hot encoding are seen as sampled from the true conditional distribution. The paper seems to confirm that one hot encoding (i.e. more labelled items rather than more labels per item) is better. Obtaining a single sample from the conditional probability is not obtaining a low quality label.

Soft labels, even if only as an additional uniform distribution over the labels have shown to increase model performances. Maybe, the authors should compare with basic synthetic soft labels (or even more complex strategies)

What is the accuracy between predicted probabilities and soft labels?



**Additional Feedback:**

NA.

**Clarity:**

Notations are very difficult to read. e.g. row 112, what is $f_{i+1}$ ? $X_{V_i}$ is a set with one element being $f_{i+1}$ ? Or is the $i$ an index and therefore $X_{V_i}$ and $X_{E_i}$ intersects ?

Figure 5 is difficult to read as it combines uncomparable metrics...

I had to read the work multiple time to get a clear idea. Many intuitions are far from the classical machine learning setup and make the paper difficult to read. A one hot encoding is described as low quality instead of sampled from the true conditional probability with large aleatoric uncertainty, etc.

**Documentation:**

NA

**Ethics:**

Yes

**Relation To Prior Work:**

Works quantifying the generalization gap with crowdsourced labels exist such as :
@inproceedings{li2013error,
  title={Error rate analysis of labeling by crowdsourcing},
  author={Li, Hongwei and Yu, Bin and Zhou, Dengyong},
  booktitle={ICML Workshop: Machine Learning Meets Crowdsourcing. Atalanta, Georgia, USA},
  year={2013},
  organization={Citeseer}
}

**Summary And Contributions:**

This paper presents a benchmark on 9 real worl datasets. The authors state that each item can be linked to aleatoric uncertainty that they assume can be inferred by collecting multiple labels.

They aim at quantifying the effect of the labelling budget be it over the number of labels per items or on the number of items labelled.

The colelcted data on all datasets consist in multiple labels per instance.

---

### Official Review · Reviewer_gHRs · 2022-07-22
**Is One Annotation Enough Review**

**Rating:** 6
**Confidence:** 3

**Strengths:**

 - 8 new datasets of images containing multiple annotations per image are provided. This allows for classification models to be trained with soft-labels. The annotations are collected from multiple human annotators.
- A variety of metrics are evaluating for both the labelling and evaluation phases, including ECE, KL divergence and accuracy
- Claims about the advantages of training with soft-labels are well supported.
- The use of a data-centric approach where the results are evaluated on a fixed model allows the differences in the labelling phase to be analysed while reducing the impact of the evaluation phase


**Weaknesses:**

- The new datasets tend to be for very specific domains, making their relevance to general image classification research questionable.
- Other than the new datasets, the paper does not contribute much new above what has already been done in previous work
- Very little detail is provided on the implementation of the benchmarked methods used for the labelling phase


**Additional Feedback:**

- On line 193 it states that having an $\hat{ACC}$ lower than 80% marks a dataset as difficult even for humans. Unless I misunderstand, $\hat{ACC}$ compares the enhanced labels to the human labels, so I am not sure how this metric relates to the difficulty of labelling the dataset for a human.


**Clarity:**

The paper is very dense and somewhat difficult to follow. It took me multiple readings to understand some basic concepts, such as what each of the phases entailed and how they were evaluated.

I think the authors have tried to include more information than can be explained adequately given the limits on the length of the paper. I think the paper would benefit from focusing on a smaller number of important concepts and results in greater detail and removing some of the less important information.

The analysis section is particularly dense and includes much information which is not explained in detail.

**Correctness:**

The datasets are labelled with multiple annotations collected by multiple human annotators, which is a much sounder method than other work which uses automatic labelling or synthetic noise.

While the general data-centric approach of using a fixed model to evaluate the contribution the label improving methods is sound, without details on the benchmarked methods it is difficult to evaluate the correctness of those methods.

**Documentation:**

The code and datasets are available, although currently a password is required. No information is provided about hosting, licensing or maintenance.


**Relation To Prior Work:**

The Related Work section describes how the datasets differ from previous datasets, except for CIFAR-10H which is included without changes. The benchmarked methods for the labelling phase are presumably not altered from previous work.

**Summary And Contributions:**

The paper presents 9 image datasets, 8 of which are new, containing multiple annotations per image to the end of training models using soft labels. It also includes benchmarks of various methods to improve the labels of multi-label datasets by measuring how the benchmarked methods affect the downstream classification task on a fixed model.

---

### Official Review · Reviewer_HJkT · 2022-07-26
**Review for paper: Is one annotation enough?**

**Rating:** 6
**Confidence:** 2
**Clarity:** In general, the paper is easy to read…

**Strengths:**

- considerable number of datasets and methods, both with various characteristics

**Weaknesses:**

- unclear what the impact of the benchmark is and what interested researchers can gain from it.
- I find there was an extensive discussion regarding ambiguous or subjective annotations in the introduction of the paper, but these aspects are not clearly discussed when benchmarking the models.
- the methods and the evaluation metrics are not well motivated

**Additional Feedback:**

- Subjectivity and ambiguity is inherent in any kind of data modality, be it natural language, images or videos. A subjective interpretation of an image does not necessarily mean that the annotator is adding noise to data. In fact, it means that we need proper ML instruments that are able to capture this subjectivity or ambiguity and reason in this space. Many times, the majority opinion is not the only one that is correct.
- The ground truth datasets in questions have been also created by humans, and as we know, humans make errors. Many times it has been shown that annotators can identify errors in existing ground truth and improve it.
- Some motivation aspects need to be given for the choice of the datasets and methods. Furthermore, it is unclear how the number of annotations was increased, what kind of annotation guidelines were used by annotators, how annotators were selected, and so on. The same comment is applicable for the evaluation metrics.
- Unfortunately, there is a lot of discussion on the subjectivity and ambiguity of annotators' labels at the beginning of the paper, but after that there is no discussion on this very important aspect of data collection and benchmarking.

**Correctness:**

There are very little details provided about the benchmark and how the evaluation was run to be very sure about this. However, I do understand that space is an issue when performing such experiments.

**Documentation:**

The details regarding the benchmarking are quite limited.

**Ethics:**

Unclear if the annotation campaigns were approved by an ethical committee.

**Relation To Prior Work:**

There is a limited discussion of approaches in the human computation and crowdsourcing areas, which are highly focusing on gathering and evaluating annotations from annotators with no or limited domain knowledge.

**Summary And Contributions:**

The paper presents a benchmark on the task of image classification which includes 9 datasets and 20 baselines. The datasets are chosen in such a way to account for different levels of difficulty and ambiguity, while the methods include supervised, semi-supervised, and self-supervised methods. Several layers of analysis are added, by looking at the amount of annotations needed, accuracy based metrics, reliability metrics, among others.

---

### Official Review · Reviewer_2hsi · 2022-07-27
**Comments from Reviewer**

**Rating:** 7
**Confidence:** 3
**Correctness:** I did not din any wrong claims in the…

**Strengths:**

The paper has a large scale study. The strengths of the paper is as follows:
1. It introduces a new set of datasets in different domains with multiple annotations which could be used for studies related to model and label uncertainty
2. It provides insights about the role of labels with a large number of experiments
3. Having different label budgets and using supervised, semi-supervised and self-supervised models
4. The collection process and the experimental setup is well documented and motivated

Overall, I think the paper makes a nice study by providing and exploiting multiple annotations.


**Weaknesses:**

One limitation is the fact that most datasets considered in this study contain fewer than 10 classes. Although it would be interesting to see if the situation changes with more fine-grained classes (e.g. 100 classes), I do not think that limits the insights greatly.

**Additional Feedback:**

No additional feedback.

**Clarity:**

It is mentioned in line 40 that the task is to improve the labels, but in my opinion, the explanation about improving label estimates could have been more detailed. Could the authors more clearly elaborate on the second step of the Labeling phase?

**Documentation:**

Details of the datasets are described in the supplementary material.

**Ethics:**

I did not find any ethical issue.

**Relation To Prior Work:**

The relation to prior works is is discussed well.

**Summary And Contributions:**

This paper runs a large scale study of the effect of high-quality labels on the performance of the models, and introduces benchmarks on the proposed dataset collected with multiple annotations to study the effect of label uncertainty. The authors particularly investigate the role of good label estimation on the actual evaluation phase, and conclude that improving the quality of the labels has a direct impact on the performance of the models (labeling phase and evaluation phase).

---

### Official Review · Reviewer_WQKr · 2022-07-27
**Useful dataset and benchmark. Writing needs work.**

**Rating:** 7
**Confidence:** 3

**Strengths:**

1. The proposed benchmark is comprised of data from a diverse set of nine existing datasets with multiple annotations. Each subset has been carefully curated and organized to contain sufficient number of annotations, with authors hiring annotators to provide additional annotations when needed.
2. The authors provide open source code to facilitate research use of the proposed benchmark. The code documentation is sufficient and clear instructions are provided on how to apply the benchmarks to additional methods.
3. Extensive experiments on 9 datasets and 20 baseline methods shed light on important findings regarding methods to refine annotations, as well as the effect of the scale of annotations.

**Weaknesses:**

1. The writing is ambigiuous with regards to the problem setting of the paper. The paper often refers to various related but distinct topics such as label noise, label ambiguity, hard vs soft ground-truth distributions synonymously, without justification or clear distinction.
2. Related to the point above, the paper does not provide a clear exposition of related works pertaining to each relevant line of work, individually. Furthermore, the need and potential impact of the proposed work is not clearly explained in the context of existing literature.
3. The writing is unclear or hard to follow in some parts of the paper. See Clarity below.
4. The use of ImageNet pre-training (i.e., abundant labeled samples) for some methods, as described in Table 2, may affect the analysis between different initialization schemes, given that CIFAR-10H, Treeversity, and Turkey datasets consist of natural images. This is not addressed in the paper.

**Additional Feedback:**

- I recommend more verbose explanations of the figures and tables presented in Section 3 for clarification.
- I recommend that the authors provide installation instructions in the codebase for a general environment, say a fresh install of Linux, where researchers are more familiar, compared to a Dockerized environment.

**Clarity:**

While there are no significant technical errors in the writing, some parts of the paper are unclear or hard to follow. This includes, but not limited to:

- Without prior knowledge in related works, it is difficult to understand what each phase of the benchmark refers to. Clarifications could be made in the third paragraph of the introduction in lines 67-81 regarding the structure of the benchmark. The following phrases are especially confusing at first glance: “initialization for an unlabeled dataset”, “describes the way the annotations are used to generate the label estimates”, “downstream classification task”.
- In Section 3, some details are unclear or hard to follow. For example, in the paragraph starting at line 197 and Figure 5, the meaning of budget is not explicitly explained—whether it refers to number of annotations, portion of images, or a mix of both. It appears to refer to number of annotations, as implied in the SL and SL+ initialization schemes, specified in lines 166-171.
- There is no explanation on how the baseline methods are used for label refinement in neither the related works (line 67) nor methods (line 154). A brief overview may help readers understand the problem setting.

**Correctness:**

The methodology used to construct the dataset as well as experiment design is overall sound. However, as mentioned in the weaknesses, the use of ImageNet labeled samples by some of the methods may invalidate the analysis on initializations.

**Documentation:**

The paper provides details on the collection, organization and hosting of the data in the main paper and supplementary material. Societal impacts are breifly discussed and licensing details are included in the code repository. The code repository provides extensive instructions to support reproducability.

**Relation To Prior Work:**

The authors discuss many prior works related to the paper. However, the organization of the related works (lines 67-81) is hard to follow, and it is difficult to identify and discern the relevant lines of work and associated literature. As mentioned in the weaknesses, it is not clearly discussed what the contribution of this paper, especially the experiments, are with regards to the previous literature.

**Summary And Contributions:**

The paper proposes a data-centric image classification benchmark designed to study "robust" methods that refine annotations from an existing (semi-) annotated dataset. The benchmark is structured in four distinct phases: (1) initialization/selection of annotations, (2) annotation refinement with a given method, (3) training a fixed model, (4) evaluation. This enables the independent study of robust methods and annotation initializations, using real labels. The proposed dataset is a curation of nine different existing image classification datasets with diverse characteristics, organized to contain sufficient annotations per sample. The authors perform extensive experiments on these datasets and 20 baseline methods to shed light on the effect of data quality (i.e. number of annotations per sample) and quantity (i.e., number of annotated sample) w.r.t. classification performance and ability to learn the true underlying distribution of data.

---

### Official Review · Reviewer_Kd8T · 2022-07-28
**Interesting work while some details need to be improved**

**Rating:** 6
**Confidence:** 2
**Correctness:** I think the claims made in the submis…
**Clarity:** The paper is well organized and easy …

**Strengths:**

1. This work provides a wide-ranged realistic benchmark with nine datasets and 20 baselines, which is a strong benchmark for the whole research community.
2. The paper is easy to follow and reads well.
3. Extensive experimental results and visualization are provided. The code is also provided, making it easy to reproduce the result.


**Weaknesses:**

It would be better if the clarity of the figure can be improved and the annotation of the figure can be much clear (e.g. Figure 6 & 7).

**Additional Feedback:**

NA

**Documentation:**

Yes, the authors open source their data and code.

**Ethics:**

No ethical concerns.

**Relation To Prior Work:**

Yes, it's well-discussed.

**Summary And Contributions:**

Authors collected nine image classification datasets with multiple annotations, which allow a realistic simulation of noise patterns. Based on these datasets, a data-centric image classification benchmarkis proposed  for noisy and ambiguous label estimation. 20 methods are evaluated as baselines and analyzed. Futhermore, this paper reveals that one annotation per image is not enough and and is ill-suited to learn the underlying ground truth distribution.

---

### Meta-Review · Area_Chair_TqvW · 2022-09-09

**Recommendation:** Accept
**Confidence:** 4

**Metareview:**

Overall, reviewers appreciate the strength of the benchmark (9 datasets in different domains with multiple annotations, 20 baseline methods) and agree that it will benefit the research community in exploring future work related to labeling budget and model and label uncertainty. Many of the weaknesses pointed out by reviewers (e.g. insufficient related work, lack of motivation for certain decisions, lack of clarity) were addressed by the authors in revisions. I recommend a spotlight presentation because the topic of label annotation quality is important and often overlooked when curating ml datasets.

---

### Decision · Program_Chairs · 2022-09-16

Accept